# The Self-Reference Effect in Metamemory and the Role of Beliefs in This Process

**DOI:** 10.3390/bs14090741

**Published:** 2024-08-25

**Authors:** Ruoyu Hou, Hui Xu, Yuanxia Gao, Weihai Tang, Xiping Liu

**Affiliations:** Faculty of Psychology, Tianjin Normal University, Tianjin 300387, China; 2210340031@stu.tjnu.edu.cn (R.H.); 2000340008@stu.tjnu.edu.cn (H.X.); 2300340007@stu.tjnu.edu.cn (Y.G.)

**Keywords:** metamemory, judgments of learning, self-referential encoding, semantic encoding, metamemory beliefs

## Abstract

Previous research has shown a clear self-reference effect in our memory. However, the question arises as to whether this effect could extend to higher cognitive domains such as metamemory. Thus, this study examined the effects of different encoding types on judgments of learning (JOLs) and explored the role of beliefs in this process. A one-way (encoding type: semantic, self-referential) within-participants design was employed in Experiment 1, which found no self-reference effect in JOLs. In Experiment 2, we manipulated participants’ beliefs to explore their effect on JOLs under different encoding strategies. The results showed that learners’ metamemory beliefs about encoding types influence JOLs. Learners who believed that self-referential and semantic encoding had the same memory effect tended to give equal JOLs to both words. However, learners who believed that self-referential encoding had a better memory effect than semantic encoding gave higher JOLs to self-referentially encoded words. The conclusions are as follows: There is no self-reference effect in JOLs, but learners’ metamemory beliefs about encoding types influence JOLs.

## 1. Introduction

The “self” is a unique mental representation, and information about oneself is often better remembered. Numerous studies showed that compared with other encoding strategies (e.g., phonemic encoding, structural encoding, semantic encoding, and other-reference), self-referential ones yield superior memory, which is called the self-reference effect [1,2]. As far back as the last century, researchers demonstrated the robust stability of this effect [2].

The experimental paradigm for the self-reference effect usually includes three tasks: an encoding task, a distraction task, and a recall test [3]. During the encoding phase, trait adjectives such as “honest” are often presented on a screen and participants assess whether the word describes their personality characteristics with a “yes” or “no” answer [1,3,4,5]. Self-referential encoding is usually compared with other encoding conditions such as semantic or other-referential encoding. Each encoding condition has its way of asking questions. Semantic encoding, for example, asks participants to determine whether a word is positive or to answer other semantic questions [4]. A distraction task follows the encoding task to avoid non-conscious rehearsal. Eventually, the researcher conducts a memory test with participants who are usually required to recall as many previously encoded words as possible in a free-form manner within a set time. Research shows that, unlike other encoding strategies, self-referential encoding uses the self, a well-elaborated and well-differentiated memory structure, which allows the material to be better organized while leaving a powerful memory trace [2,6].

In recent years, researchers found that the self-reference effect can be extended to many other cognitive processes such as visual perception [7], attention [8], executive function [9], language processing [10], and problem-solving [11], showing great scalability. Unfortunately, although self-reference effects are present in human memory and other cognitive systems, among the studies we know of, little attention has been paid to whether these effects can be generalized to more advanced cognitive activities like metamemory. As such, the present study fills a gap in the literature in this area. Through two experiments, we explored for the first time whether there is a self-reference effect in item memory monitoring and investigated the role of beliefs in this process.

### 1.1. Judgments of Learning and Theoretical Explanations Thereof

Metamemory refers to metacognition about memory, including memory monitoring and control. Learners make learning decisions and regulate learning strategies, which significantly affect the efficiency and effectiveness of learning, based on memory monitoring [12,13,14]. Judgments of learning (JOLs), a typical type of memory monitoring, refer to learners’ predictions of how well they will do on subsequent retrieval tests for items they have studied [15]. In studies on JOLs, participants learn items and then make JOLs for them. When making JOLs, participants select numerical values representing their likelihood of retrieval on a continuous scale of 0–100 (0 = definitely would not retrieve, 100 = definitely would retrieve) [16,17,18]. They then complete a memory test. By analyzing JOLs, researchers can understand participants’ self-confidence in memory tasks. The closer JOLs are to actual retrieval performance in the final test, the more accurate their memory monitoring is.

How do people make JOLs? According to the cue-utilization account, JOLs are processes through which learners draw inferences from various cues [19]. In general, task-centered cues are emphasized more than learner-centered ones [20]. Inherent characteristics of study materials that have been demonstrated to affect JOLs are word frequency [21], relatedness of word pairs [22], font size [23,24], concreteness [25], and animacy of items [26,27]. In addition, some learning conditions also affect JOLs, such as learning time [28] and the interval between learning and testing [29]. Koriat refers to the inherent characteristics of learning materials as intrinsic cues and to learning conditions as extrinsic cues, arguing that both these cues are important in making JOLs. Finally, the internal, subjective indicators of learners, known as mnemonic cues, can also influence JOLs [19]. The dual-basis model concretely explains how individuals use the above cues to make metacognitive judgments. The model proposes that people may rely on two processing systems when making JOLs: One is theory-based, in which people make inferences from their beliefs about memory. The other is experience-based, in which people depend on their subjective experience of performing a task, such as processing fluency, which refers to the difficulty of an item in mental processing [30].

Analytic-processing theory further refines the cognitive process of memory monitoring. The theory suggests that when people monitor their learning, they use their own metamemory beliefs as cues to analyze and speculate to reduce the uncertainty of subsequent memory [31,32]. Numerous studies confirmed the significant role of metamemory beliefs in the process of making JOLs [22,24,25,33,34,35]. For example, Chen et al. [36] examined how metamemory beliefs about processing fluency influence the effects of font size by manipulating participants’ beliefs. They informed one group that large fonts can be processed more smoothly than small ones and that simple processes often lead to better memory outcomes. The other group was told that fluency has nothing to do with memory. The results indicated that the former group made higher JOLs for large fonts than small fonts, but no difference in JOLs was observed in the latter group. This suggests that learners are influenced by their beliefs when producing JOLs for words in different font sizes. Ikeda [34] explored whether beliefs contribute to the impact of achievement goals on JOLs, showing that individuals used beliefs regarding achievement goals when predicting their future test performance. These results provide empirical evidence for the analyzing–processing theory.

### 1.2. Self-Reference and Judgments of Learning

From the cue-utilization account, encoding conditions are the extrinsic cues that learners rely on when making JOLs [19]. However, while self-reference is a typical and effective encoding strategy, little attention has been paid to its effects on JOLs.

Previous studies investigated the differential performance of self-generation and other-generation in terms of item and source memory, and both types of memory monitoring. For example, in Experiment 2 of Carroll et al. [37], participants were presented with action descriptions and required to perform four responses (act themselves, witness the experimenter act, imagine themselves acting, imagine the experimenter acting). They then predicted the likelihood that they would be able to recall these actions (JOLs) and their sources (judgments of source, JOSs) a week later. The results showed that participants rated self-focused items as more likely to be remembered than other-focused ones, and the magnitude of JOS favored self-performed actions over other types. Similarly, in a study by Carroll et al. [38], two groups of participants completed different tasks: One group generated words by themselves, and the other group listened to their peers generating words. After that, both groups made JOLs and JOSs. The results showed that participants perceived self-generated words as having a higher likelihood of item and source recognition than peer-generated words. Although the two studies did not use a standard experimental self-reference paradigm, their results suggest a self-reference effect in metamemory; specifically, people may make higher metamemory judgments for items associated with the self.

To our knowledge, only one experiment has explored the difference between self-referential and semantic encoding in metamemory. Pereira et al. [39] studied the effects of emotional valence and encoding style on source memory and the monitoring thereof. In Experiment 2, participants learned and made immediate JOSs for words with different emotional valence (positive, negative, and neutral). During the studying phase, half the words were self-referentially encoded (to assess whether these items were relevant to participants’ characteristics), and the other half were semantically encoded (to evaluate whether the items were regularly used). Immediately after each word was encoded, participants predicted the likelihood of recognizing the source of the word using a six-point rating scale (higher points mean more confidence). The results suggested that participants gave higher JOSs for the self-referentially encoded words than the semantically encoded ones, indicating that encoding strategies can affect source memory monitoring. Item and source memory monitoring are highly associated in some findings, and researchers speculated that learners may use the same cues for prediction in both tasks [37,38]. However, it was also found that JOLs and JOSs appear distinct and that there may be different monitoring processes between these two judgments [40]. Therefore, researchers argued that specific types of metamemory judgments must be considered when discussing the determinants of metamemory [41]. However, since Pereira et al. [39] did not investigate whether self-reference would have the same effect in predicting item memory, the existence of the self-reference effect in metamemory and its cognitive process still needs further exploration.

### 1.3. The Present Study

In summary, although the self-reference effect is robustly stable in memory [2], little is known about whether it also exists in metamemory. In addition, no research has investigated people’s metamemory beliefs about the self-reference effect. Therefore, it is not clear whether people believe in the existence of the self-reference effect, much less whether they apply their metamemory beliefs about encoding strategies to their metamemory judgments. Investigating this issue will contribute to a better understanding of the characteristics of metamemory when people use different encoding modalities and reveal the role of metamemory beliefs in metamemory judgments.

Thus, we examined whether there is a self-reference effect in metamemory (specifically referring to item memory monitoring in this study) and the role of beliefs therein. By combining the standard self-reference paradigm with that of JOLs, we investigated how self-referential and semantic encoding affects JOLs. Given the stability of the self-reference effect in recall performance, we anticipated that it would still occur after making JOLs. (In other words, participants would recall more words in the self-referential encoding condition than in the semantic encoding condition.) In terms of JOLs, this study expected to obtain similar results to Pereira et al. [39], namely that encoding strategies would also affect JOLs, and self-referential encoded items would receive higher JOLs than those semantically encoded. In Experiment 2, we manipulated participants’ beliefs to investigate their performance on JOLs after completing different encoding tasks under different belief conditions to evaluate the analytical-processing theory. As mentioned, there is an implicit condition in analytical-processing theory that if people hold certain metamemory beliefs, they will apply them to JOLs. In other words, if people assume that using a self-referential encoding strategy will lead to better recall performance, they will give higher JOLs to self-referentially encoded items. Conversely, if they believe that encoding conditions do not affect retrieval performance, there should be no significant difference in JOLs under the two different encoding conditions.

## 2. Experiment 1

### 2.1. Materials and Methods

#### 2.1.1. Participants and Design

Before participant recruitment, the sample size was determined a priori using G*power 3.1.9.7 statistical software [42], with power = 0.80, α = 0.05, and a medium effect size of 0.5 to detect differences in memory monitoring performance across encoding types via a one-way (encoding type: semantic, self-referential) within-participants design. The calculation showed that 34 participants would be sufficient. Therefore, 34 undergraduate and graduate students (18 females) with normal vision (including vision corrected to normal) with an average age of 19.76 years (*SD* = 1.72) were enrolled for Experiment 1. No participants had previously participated in similar experiments and all were rewarded after the experiment of this study. As the two dependent variables, each participant’s recall performance (proportion of correctly recalled items) and JOLs were measured.

#### 2.1.2. Materials

In total, 32 words were selected from the 168 Chinese double-character trait adjectives used by Zhang et al. [43], and 26 students (*M*_age_ = 22.08, *SD* = 2.10; 19 females) were recruited to rate the difficulty and familiarity of the words on a 5-point scale. All the adjectives were rated as between 1.54 and 2.35 in difficulty (1 = very easy, 5 = very difficult) and 3.96 to 4.69 in familiarity (1 = very unfamiliar, 5 = very familiar). The 32 trait adjectives consisted of 16 positive and 16 negative adjectives. The valences of the words were evaluated by 22 students (*M*_age_ = 23.36, *SD* = 3.17; 16 females) with scores ranging from 1.59 to 4.36 (1 = very positive, 5 = very negative). No extreme valence scores were reported.

Subsequently, the 32 words were divided into 2 blocks of 16 words in each block (each including 8 positive and 8 negative adjectives). The words in the two blocks were matched in terms of difficulty, familiarity, and valence, and there were no differences in any of the three dimensions (*t*(25.11)_difficulty_ = 0.673, *p* = 0.507; *t*(30)_familiarity_ = 0.65, *p* = 0.521; *t*(30)_valence_ = −0.08, *p* = 0.939). In addition, four unselected words with similar difficulty and familiarity as the selected words were selected from the experimental materials of Zhang et al. [43] for the practice phases of the two blocks (each block contains a positive and a negative word).

#### 2.1.3. Procedure

Learning, distraction, and recall tasks were performed sequentially. Participants had to read and understand the instructions, and successfully pass the practice phase before entering the formal experiment.

As Figure 1 shows, the learning task required participants to encode each word and make JOLs for them. There were two encoding tasks, each contained within a block. The 16 adjectives in each block were presented one by one in random order. Participants had 5 s to complete the encoding process as instructed. Half the participants engaged in self-referential encoding followed by semantic encoding, while the other half did the reverse. The order effects were balanced by the ABBA method. Different questions were displayed on the screen for different encoding conditions. For self-reference, the question was: “Does this adjective describe you?” The question for semantic encoding was: “Is this adjective a positive word?” Underneath these questions, the options “Yes” or “No” were provided. Participants had 5 s to respond by pressing “Q” or “W” on the keyboard. If there was still time left after completion, the screen continued to present the word and question. Following the encoding task, a rating bar ranging from 0 to 100 appeared below each word, requiring participants to make a JOL for the word (to predict how likely they were to recall it on a subsequent test: 0 = definitely would not recall, 100 = definitely would recall). Participants clicked on their estimated position along the rating bar and then on the “Submit” button to jump to the next word learning interface.

After learning all the items in all the blocks, participants were instructed to perform a one-minute mathematical task of subtracting three consecutively from a three-digit number. Finally, a test was conducted wherein participants were required to freely recall as many words as possible in 3 min and report them aloud.

### 2.2. Results

The data were analyzed using SPSS 27.0. Table 1 provides the descriptive statistics of learners’ recall performance and JOLs under different encoding type conditions.

#### 2.2.1. Recall Performance

A paired samples T-test was conducted on the recall accuracy in the two encoding-type conditions. The results, shown in Figure 2a, indicate that participants’ recall performance under the self-referential encoding condition was significantly higher than that in the semantic encoding condition (*t*(33) = −4.33, *p* < 0.001, *d* = −0.742). This demonstrates a strong self-reference effect.

#### 2.2.2. JOLs

We analyzed JOLs (Figure 2b) with a paired sample T-test. Participants’ JOLs did not differ significantly between the encoding conditions (*t*(33) = −0.56, *p* = 0.581). This indicates that participants had equal self-confidence in both conditions, and there was no self-reference effect in JOLs.

### 2.3. Discussion

The results of Experiment 1 showed that although participants demonstrated the self-reference effect in item memory, there was no significant difference in JOLs between the self-referential and semantic encoding conditions. In other words, the self-reference effect was not found in metamemory. This contrasted with the results reported by Pereira et al. [39], who found that JOSs were higher in the self-referential encoding condition than in the regular semantic encoding condition, suggesting a self-reference effect in source memory monitoring. The different results of the two studies may be attributed to differences in the processes of item memory and source memory monitoring.

## 3. Experiment 2

In Experiment 1, the encoding types did not affect JOLs. There are two possible reasons: either the participants did not realize that self-referential encoding is better for memory than semantic encoding, or they held the above belief but would not use it as a cue for conducting JOLs. Therefore, we manipulated participants’ beliefs in Experiment 2 to explore how beliefs affected JOLs under different encoding strategies. The present study anticipated an interaction between beliefs and encoding conditions. Participants who believed that self-referential encoding has the same memory effect as semantic encoding would assign equal JOLs to words from both encoding types, but those who held the belief that self-referential encoding is more favorable to memory would assign higher JOLs to self-referentially encoded words.

### 3.1. Materials and Methods

#### 3.1.1. Participants and Design

In total, 60 students participated in Experiment 2. We randomized them into 2 groups of 30. One group (*M*_age_ = 20.27, *SD* = 1.64; 20 females) was told that “self-referential and semantic encoding have the same memory effect” (hereafter referred to as “SRE = SEM”). The other group (*M*_age_ = 20.13, *SD* = 2.54; 19 females) was told that “self-referential encoding has a better memory effect than semantic encoding” (hereafter referred to as “SRE > SEM”). At the end of the experiment, all participants were provided with a small reward. G*Power3.1.9.7 statistical software [42] calculated that a total sample size of at least 34 people would reach the power level of 0.8 at an alpha significance level of 0.05 and medium effect size (*f* = 0.25) for this 2 × 2 mixed design with belief (SRE = SEM or SRE > SEM) manipulated as a between-participants factor and encoding type (semantic or self-referential) as a within-participants factor. This indicated that the sample size was adequate. As in Experiment 1, the dependent variables in Experiment 2 also recalled performance and JOLs. Four participants were excluded for failing to meet the data requirements of this experiment (see the Results Section 3.2 for the reasons). The mean age of the remaining 56 participants (38 females) was 20.21 years, *SD* = 2.18.

#### 3.1.2. Materials

The materials were the same as those in Experiment 1.

#### 3.1.3. Procedure

The procedure for Experiment 2 was similar to that of Experiment 1. The difference was that a belief manipulation was added before the learning phase. Participants first read a piece of message aloud and did not enter the learning phase until they fully understood the meaning of the material. The SRE = SEM group was given the information that studies have shown that thinking about whether an adjective is a positive word has the same memorization effect as thinking about whether the adjective describes you. The SRE > SEM group was informed that studies have shown that thinking about whether an adjective describes you is better for memorization than thinking about whether the adjective is positive or not. After the belief manipulation, participants successively completed the learning, distraction, and testing tasks as in Experiment 1. At the end of the test, participants were asked to recall the material they had read and to rate how much they trusted the information when they read it on a seven-point scale (1 = not at all, 7 = completely).

### 3.2. Results

All participants recalled the material they had read before the learning phase. Three participants in the SRE = SEM group and one in the SRE > SEM group who did not trust the reading material were excluded (trust rating less than three). The efficiency of belief manipulation in the SRE = SEM group was 90.00% (*M*_trust rating_ = 4.59, *SD* = 0.93). The efficiency of belief manipulation in the SRE > SEM group was 96.67% (*M*_trust rating_ = 5.07, *SD* = 1.22). There was no significant difference in the level of trust between the two groups (*t*(54) = −1.63, *p* = 0.109). Data from the remaining 56 participants were analyzed using SPSS 27.0. Table 2 shows the descriptive statistics of recall performance and JOLs for participants with two different beliefs under different encoding conditions.

#### 3.2.1. Recall Performance

A 2 × 2 mixed ANOVA conducted on recall performance (see Figure 3a) confirmed the significant main effect of encoding type (*F*(1, 54) = 21.16, *p* < 0.001, *η_p_*^2^ = 0.282): a strong recall advantage was demonstrated for self-referentially encoded items, while no main effect was found for belief, *F*(1, 54) = 0.01, *p* = 0.933, and no interaction was found, *F*(1, 54) = 0.05, *p* = 0.824. The simple effect analysis showed that regardless of participants’ beliefs, the recall accuracy of self-referential encoding was significantly higher than that of semantic encoding (*F*(1, 54)_SRE = SEM_ = 9.25, *p* = 0.004, *η_p_*^2^ = 0.145; *F*(1, 54)_SRE > SEM_ = 12.07, *p* = 0.001, *η_p_*^2^ = 0.185). Furthermore, no significant difference was found in the recall accuracy of the same encoding type under different belief conditions (*F*(1, 54)_SEM_ = 0.06, *p* = 0.812; *F*(1, 54)_SRE_ = 0.01, *p =* 0.917).

#### 3.2.2. JOLs

Using JOLs as the dependent variable, a 2 (belief: SRE = SEM, SRE > SEM) × 2 (encoding type: semantic, self-referential) mixed ANOVA was performed. Figure 3b shows the results. A marginal main effect of belief was found, *F*(1, 54) = 3.49, *p* = 0.067, *η_p_*^2^ = 0.061, such that participants gave higher JOLs under the SRE > SEM belief condition than under the SRE = SEM one. No main effect of encoding type was found, *F*(1, 54) = 2.47, *p* = 0.122, and the interaction between belief and encoding type was also not significant, *F*(1, 54) = 2.28, *p* = 0.137. A simple effect analysis showed no significant difference in JOLs for the SRE = SEM group between the two encoding types, *F*(1, 54) = 0.00, *p* = 0.967; however, the SRE > SEM group provided higher JOLs for self-referentially encoded words than they did for semantically encoded words, *F*(1, 54) = 4.92, *p* = 0.031, *η_p_*^2^ = 0.083. In addition, when participants engaged in semantic encoding, there was no significant difference in JOLs between the two groups, *F*(1, 54) = 1.62, *p* = 0.209. However, when participants performed self-referential encoding, JOLs were significantly higher in the SRE > SEM group than in the SRE = SEM group, *F*(1, 54) = 5.44, *p* = 0.023, *η_p_*^2^ = 0.092. This suggested that participants’ beliefs influenced their prediction of recall performance under different encoding-type conditions.

### 3.3. Discussion

The stability of the self-reference effect was again verified, as it was not affected by differences in beliefs. Furthermore, Experiment 2 demonstrated that participants’ metamemory beliefs about encoding strategies influenced their JOLs. As expected, when learners believed that using self-referential encoding would lead to a better recall outcome, a self-reference effect in JOLs was found. In contrast, when learners believed that the two encoding types would produce the same recall effects, they assigned equal JOLs to the words under the two encoding conditions; thus, no self-reference effect in JOLs was found. These results support the analytic-processing theory.

## 4. General Discussion

This study examined whether a self-reference effect in metamemory exists and the role of metamemory beliefs therein through two experiments. Experiment 1 found that when learners were not given belief cues, they assigned equal JOLs for self-referentially and semantically encoded words, exhibiting no self-reference effect. However, Experiment 2 indicated that when participants were given a relevant metamemory belief cue, they applied it to the process of JOLs. Specifically, when participants believed that self-referential encoding is more favorable to memory, they gave significantly higher JOLs for self-referentially encoded words than for semantically encoded ones. Conversely, when they believed that different encoding strategies make no difference in memory, they gave equal JOLs for words in both conditions. The results revealed that individuals’ metamemory beliefs about encoding types are important cues for their production of JOLs. When participants held metamemory beliefs about the self-reference effect, their metamemory was populated with a self-reference effect.

### 4.1. Self-Reference Effect in Metamemory

In both experiments, recall accuracy for self-referential encoding was much higher than that for semantic encoding, regardless of whether participants’ beliefs were manipulated. This illustrates the strong robustness of the self-reference effect in memory, which corroborates with a large body of previous research [2,3]. However, while many studies showed that the self-reference effect can be extended to other cognitive processes, Experiment 1 found that this extension may not be fully applicable to more advanced metamemory.

According to the cue-utilization account, encoding methods are among the external cues that can be utilized [19]. Pereira et al. [39] explored whether different encoding strategies affect learners’ JOSs in Experiment 2, finding that they produced significantly higher JOSs for self-referentially encoded words than for semantically encoded ones. This exemplifies the existence of the self-reference effect in metamemory from the perspective of source memory monitoring. The present study focused on whether the self-reference effect in metamemory is also expressed in item memory. It, however, did not obtain similar results to Pereira et al. [39]. On one hand, in Pereira et al. [39], the encoding process was self-paced, which may have led to longer learning times for self-referentially encoded words so that learners may have been prone to overestimate source memory performance. On the other hand, unlike the affective norms and old–new recognition patterns used by Pereira et al. [39], this study used trait adjectives as experimental material and conducted the memory test in a free recall format. Thus, the differences in results may also reflect the fact that the effect of encoding type on metamemory is moderated by the type of material and test.

Furthermore, Kelly [40] examined the effects of different modes (real or imagined) and proportions of item presentation on JOLs and JOSs. That study found that, unlike JOLs, participants were not sensitive to the proportions when making JOSs, and their reality monitoring performance did not differ depending on the number of times real versus imagined items were presented. This suggests that the same cues may have different levels of influence on JOLs and JOSs. In other words, learners may use different cues when performing JOLs and JOSs, and the two metamemory processes may diverge. Therefore, the differences between the results of the present study and those of Pereira et al. [39] may also be related to the fact that encoding methods can have different effects on JOLs and JOSs. As Schaper et al. [41] suggest, the results of our study reveal that different types of metamemory judgments should be considered when exploring the presence of the self-reference effect in metamemory.

### 4.2. Role of Metamemory Beliefs on JOLs under Different Encoding Conditions

The cue-utilization account argued that metamemory judgments, including JOLs, are inferential processes and that people rely on multiple cues to predict the likelihood of successfully recalling an item on a subsequent test [19]. This study found no self-reference effect in JOLs under natural conditions, but when learners held metamemory beliefs about the self-reference effect, they applied such belief cues to making JOLs and provided a higher number thereof for self-referentially encoded words. These results support the analytic-processing theory. As mentioned, the analytic-processing theory suggests that individuals predict their subsequent recall performance based on their beliefs and such predictions do not readily change with learning experience [31,32].

Researchers argued that the “gold standard” for testing the role of beliefs in JOLs is whether they moderate the effect of a variable on JOLs [27,44,45]. In Experiment 2 of this study, participants who held different beliefs performed differently on JOLs for words with different encoding strategies. Those who believed in the self-reference effect predicted that self-referentially encoded words would be remembered better, while those who believed that encoding type did not affect recall performance predicted the same recall performance for the two encoding types. This suggests that beliefs play a moderating role in encoding types affecting JOLs. It is no coincidence that previous studies found similar patterns of memory monitoring in the effects of variables such as relatedness of word pairs [22], material concreteness [25], font size [24], word frequency [46], and achievement goals [34] on JOLs. For example, Witherby and Tauber [25] noted that people believed that concrete words (e.g., “table”) are easier to remember than abstract words (e.g., “loyalty”), and so applied this belief to JOLs.

It is important to note, however, that in Experiment 2, the results of the ANOVA on the JOLs indicated that the interaction between belief and encoding type did not reach statistical significance. The findings that metamemory beliefs about encoding styles affected participants’ JOLs were obtained through post hoc comparisons. Consequently, these results need to be treated with statistical caution. In addition, beliefs do not seem to fully explain the effect of a variable on JOLs. Some studies indicated that beliefs do not always work [20,27,35,47]. For example, Witherby et al. [20] found that learners believed that memory is better when people’s emotions are congruent with the valence of the learning items, but did not apply this belief to JOLs. The question of why beliefs only work on some cues needs further exploration. Furthermore, analytic-processing theory does not exclude the contribution of other factors such as processing fluency during learning. Yang et al. [35] analyzed previous research and concluded that, although beliefs generally influence JOLs, in some cases, processing fluency also affects JOLs concurrently with beliefs. Therefore, beliefs may not be the sole factor at play. 

### 4.3. Limitations and Future Research

Based on the comparison with previous studies, this research speculates that the different performances of semantic and self-referential encoding in metamemory may be conditioned by several other factors such as the type of material (trait adjectives, nouns, stories, pictures, etc.), test mode (recognition, recall), time of encoding (time-limited, time-unlimited), and type of metamemory judgments (JOSs, JOLs). However, the specific manifestations of these need further study. In recent years, exploring how learners integrate multiple cues for memory monitoring has gradually become a research trend in metamemory, as opposed to examining the effect of a single cue on metamemory [20,48,49]. This trend should be capitalized on to examine how learners integrate different cues associated with encoding types to make JOLs.

Compared to the difference between self-referential and semantic encoding, the self-reference effect is weakened when the comparison condition involves others (i.e., self-referential versus other-referential) [2]. In addition, the effect of other-referential encoding on memory is also moderated by the participant’s relationship with this person [2,43]. However, since it has not been verified whether self-referential and other-referential encoding have different effects on metamemory, this question could be explored in the future to deepen understanding of this area.

Finally, there are third obvious limitations of this study. First, as mentioned above, although Experiment 2 found that participants’ metamemory beliefs affected their JOLs, this result was obtained from a simple effects analysis conducted after the interaction was found to be non-significant. Therefore, future experimental validation is needed to ensure the reproducibility of this result and to draw scientifically rigorous conclusions. Second, although this study demonstrated that beliefs do play a role in the effect of encoding strategies on JOLs, only one validation method (belief manipulation) was used. Other methods include belief questionnaires [25,26], pre-study JOLs [25,31], and learner–observer tasks [50,51]. Given that each method has its strengths and limitations [35], different measures should be used in the future to verify the reliability of our results. Third, this study did not isolate the role of processing fluency. Furthermore, the effect of encoding style on JOLs may also be influenced by factors such as the level of effort during encoding and the level of mental arousal. Therefore, it has yet to be verified whether beliefs play a unique role herein.

## 5. Conclusions

There is no self-reference effect in JOLs, but learners’ metamemory beliefs about encoding types influence JOLs. Learners who believe that self-referential and semantic encoding have the same memory effect tend to give equal JOLs to both words. However, learners who believe that self-referential encoding has a better memory effect than semantic encoding gave higher JOLs to self-referentially encoded words.

## Figures and Tables

**Figure 1 behavsci-14-00741-f001:**
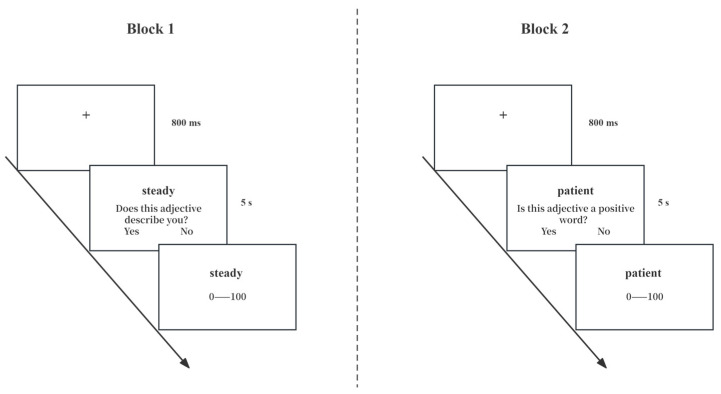
Procedures of the learning phase in Experiment 1.

**Figure 2 behavsci-14-00741-f002:**
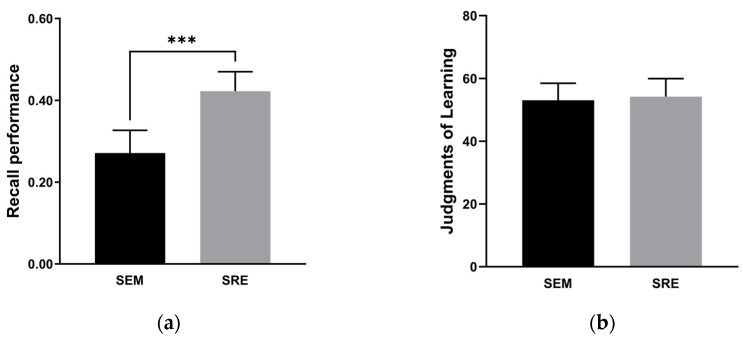
(**a**) Recall performance for each encoding type for Experiment 1; (**b**) JOLs for each encoding type for Experiment 1. ***: *p* < 0.001. The error bars are confidence intervals.

**Figure 3 behavsci-14-00741-f003:**
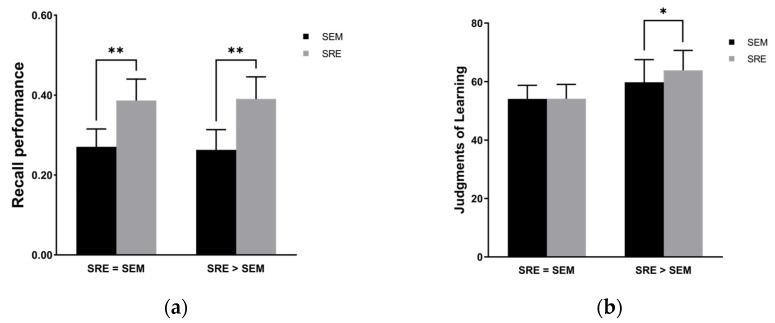
(**a**) Recall performance for each encoding type and belief for Experiment 2; (**b**) JOLs for each encoding type and belief for Experiment 2. *: *p* < 0.05; **: *p* < 0.01. The error bars are confidence intervals.

**Table 1 behavsci-14-00741-t001:** Mean Recall Performance and JOLs (*M* ± *SD*) by Encoding Type in Experiment 1.

Encoding Type	Recall Performance	JOLs
SEM ^1^ (*n* = 34)	0.27 ± 0.16	53.03 ± 15.61
SRE ^2^ (*n* = 34)	0.42 ± 0.14	54.20 ± 16.39

^1^ semantic encoding; ^2^ self-referential encoding.

**Table 2 behavsci-14-00741-t002:** Mean Recall Performance and JOLs (*M* ± *SD*) by Belief and Encoding Type in Experiment 2.

Belief	Recall Performance		JOLs	
	SEM	SRE	SEM	SRE
SRE = SEM (*n* = 27)	0.27 ± 0.11	0.39 ± 0.14	54.06 ± 11.82	54.14 ± 12.37
SRE > SEM (*n* = 29)	0.26 ± 0.13	0.39 ± 0.15	59.78 ± 20.37	63.84 ± 17.98

## Data Availability

The raw data supporting the conclusions of this article will be made available by the authors on request.

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
