# Peer review of "The Self-Reference Effect in Metamemory and the Role of Beliefs in This Process"

_behavsci, 2024, doi:10.3390/bs14090741_

Round 1

Reviewer 1 Report

Comments and Suggestions for Authors

This manuscript is a clear and concise description of two experiments designed to explore metamemory predictions of learning related to potential differences between self-referential and semantic memory performance. The authors compare metamemory judgements in the presence or absence of an influencing belief manipulation. I appreciate the supporting efforts in selecting a matched panel of words for testing and consideration of individuals' belief of the belief manipulation. Overall, this study is well designed and executed. I have only a few minor comments.

Line 220 'After learning all the items,'...does the testing occur after each block or after all blocks only? This could be more clear.

Line 270, it is not clear how you concluded that you needed the same sample size (34) in expt 1 and expt 2 when you expanded your comparisons to a 2x2 analysis in expt 2. It may be that the effect of belief manipulation was marginal because your sample size was not large enough. I think it would be reasonable to state that.

Fig 3b, could add bracket showing significant comparison: 'when participants performed self-referential encoding, JOLs were significantly higher in the SRE > SEM group than in the SRE = SEM group, F(1, 54)=5.44, p=0.023, η2=0.09'

Line 423 The following statement is contradictory: '...although beliefs always play a role in JOLs in most cases, in a few instances, ...'

Author Response

Thank you very much for your detailed review and valuable feedback on our manuscript. Your suggestions have been instrumental in improving the quality and clarity of our research. We have thoroughly revised the manuscript following your comments. Please find our detailed responses below, along with the corresponding revisions in the re-submitted files.

Comments 1: Line 220 'After learning all the items,'...does the testing occur after each block or after all blocks only? This could be more clear.

Response 1: Thank you for pointing this out. We agree with this comment. Therefore, we have changed ‘After learning all the items,’ to ‘After learning all the items in all the blocks,’ (Page 5, line 219).

Comments 2: Line 270, it is not clear how you concluded that you needed the same sample size (34) in expt 1 and expt 2 when you expanded your comparisons to a 2x2 analysis in expt 2. It may be that the effect of belief manipulation was marginal because your sample size was not large enough. I think it would be reasonable to state that.

Response 2: Thank you for pointing this out. We reviewed the calculation process for Experiment 1 (with power = 0.80, α = 0.05, and a medium effect size of 0.5) and Experiment 2 (with power = 0.80, α = 0.05, and a medium effect size of 0.25) using the G*Power 3.1.9.7 statistical software. We have confirmed that the previously calculated sample sizes for both Experiment 1 and Experiment 2 were correct. Regarding Experiment 2, we were particularly cautious about the sample size provided by the software. Consequently, we recruited 60 participants (30 participants for each of the two belief groups) to achieve a larger sample size and enhance the reliability of our results.

Comments 3: Fig 3b, could add bracket showing significant comparison: 'when participants performed self-referential encoding, JOLs were significantly higher in the SRE > SEM group than in the SRE = SEM group, F(1, 54)=5.44, p=0.023, η2=0.09'.

Response 3: Thank you for pointing this out. We apologize for any misunderstanding. Did you mean that we should include in the figure caption the statement 'When participants performed self-referential encoding, JOLs were significantly higher in the SRE > SEM group than in the SRE = SEM group, F(1, 54)=5.44, p=0.023, η2=0.09' to show the significance comparison? If so, we will add this statement to the figure caption.

Comments 4: Line 423 The following statement is contradictory: '...although beliefs always play a role in JOLs in most cases, in a few instances, ...'

Response 4: Thank you for pointing out the error. To clarify the meaning, we have modified the sentences to: ‘Yang et al. [35] analyzed previous research and concluded that, although beliefs generally influence JOLs, in some cases, processing fluency also affects JOLs concurrently with beliefs. Therefore, beliefs may not be the sole factor at play.’ (Page 10, lines 425-428).

Reviewer 2 Report

Comments and Suggestions for Authors

Hou et al. investigate the self-reference effect in the domain of metamemory and how the effect is impacted by beliefs. In Experiment 1, the authors found that trait adjectives were better recalled when participants were asked to determine whether each adjective described themselves, compared to when participants were asked to determine whether the adjectives had a positive or negative valence. This replicated previous work demonstrating the self-referencing effect on memory. However, judgments of learning (JOLs) did not differ between these conditions, suggesting a null effect on metamemory. In Experiment 2, the authors manipulated participants’ beliefs by telling them that self-referencing either has a positive or no effect on memory. The authors report a self-referencing effect of memory again in Experiment 2. Here, the authors report that JOLs were higher for self-referencing information in participants who were told that self-referencing benefits memory. The authors interpret this as a self-referencing effect on JOLs that is dependent on individuals’ beliefs about memory processes.

I thought that this paper was well-written, and the authors do a nice job of introducing and discussing these topics. Unfortunately, however, I have major concerns about how the results were analyzed and interpreted.

My primary criticism is that there is, in my view, extremely weak (if any) evidence of an impact on JOLs in Experiment 2. The authors report that there were no significant main effects for JOLs (the main effect of belief was "marginal" but not significant), and no significant interaction. Given all these null effects, follow-up post-hoc analyses would generally be considered unjustified. However, the authors go on to analyze post-hoc comparisons anyway, and find that JOLs were higher for self-referencing words than semantically encoded works for the SRE > SEM group, and that self-referencing JOLs were higher in the SRE > SEM group than the SRE = SEM group. As I mentioned, in my view these tests shouldn’t have been performed in the first place, given that there was no interaction, and not even any significant main effects. At the very least, the researchers should emphasize extreme caution in interpreting these effects. However, the tenuousness of the effects are never discussed, and are not mentioned at all, in the manuscript.

I have little confidence that the JOL results of Experiment 2 are “real.” Given the tenuousness of the findings, I see a couple of options for the authors. The first option would be to simply pare back the claims and not include the post-hoc comparisons and conclude that there is no evidence for an effect of beliefs based on the lack of an interaction or main effects, or at least acknowledge that the evidence is very weak and requires more work to make any firm conclusions. The second option I suggest would be that the authors perform a power analysis based on the Experiment 2 results they have here to get a better estimate of the sample size needed, and then repeat Experiment 2 with a new sample (ideally after a pre-registration) to see if the effects hold. I think the second option would be best. As of a result of these concerns I must recommend rejection of the paper.

More minor comments:

·         The error bars are very large relative to the difference between conditions, and in fact it’s difficult to believe that there are any significant differences between JOLs in Figure 3 (the error bars are almost entirely overlapping). I’m guessing the error bars are raw standard deviations, which are not particularly helpful when visually assessing significance. I’d highly recommend they use confidence intervals instead, preferably with a within-subject correction (e.g. Loftus-Masson). In any case, the authors need to specify what the error bars are in the figure caption.

·         The authors say for both recall performance and JOLs that they performed repeated-measures ANOVAs (lines 306, 320), which would mean that both factors were within-subject, when they should have performed mixed ANOVAs with belief as a between-subject factor and encoding type within-subject. Presumably this was a labeling error and the authors did not treat both variables as within-subject.

·         The 1 and 2 footnotes for Table 1 should be switched, I believe (SEM should correspond to semantic encoding).

·         I was a little confused in the abstract when the authors called the self-reference effect a memory monitoring process (lines 8-9), but it seems to me that the effect itself doesn’t necessarily have anything to do with monitoring per se (although the authors are investigating monitoring in the context of effects of self-referencing on metamemory).

·         It seems odd that the power analyses of Experiments 1 and 2 would calculate that the exact same number of participants (34; lines 175 and 270) was needed for both experiments given that Experiment 2 had a between-subjects manipulation.

Author Response

Thank you very much for your detailed review and valuable feedback on our manuscript. Your suggestions have been instrumental in improving the quality and clarity of our research. We have thoroughly revised the manuscript following your comments. Please find our detailed responses below, along with the corresponding revisions in the re-submitted files.

Comments 1: My primary criticism is that there is, in my view, extremely weak (if any) evidence of an impact on JOLs in Experiment 2. The authors report that there were no significant main effects for JOLs (the main effect of belief was "marginal" but not significant), and no significant interaction. Given all these null effects, follow-up post-hoc analyses would generally be considered unjustified. However, the authors go on to analyze post-hoc comparisons anyway, and find that JOLs were higher for self-referencing words than semantically encoded works for the SRE > SEM group, and that self-referencing JOLs were higher in the SRE > SEM group than the SRE = SEM group. As I mentioned, in my view these tests shouldn’t have been performed in the first place, given that there was no interaction, and not even any significant main effects. At the very least, the researchers should emphasize extreme caution in interpreting these effects. However, the tenuousness of the effects are never discussed, and are not mentioned at all, in the manuscript.

I have little confidence that the JOL results of Experiment 2 are “real.” Given the tenuousness of the findings, I see a couple of options for the authors. The first option would be to simply pare back the claims and not include the post-hoc comparisons and conclude that there is no evidence for an effect of beliefs based on the lack of an interaction or main effects, or at least acknowledge that the evidence is very weak and requires more work to make any firm conclusions. The second option I suggest would be that the authors perform a power analysis based on the Experiment 2 results they have here to get a better estimate of the sample size needed, and then repeat Experiment 2 with a new sample (ideally after a pre-registration) to see if the effects hold. I think the second option would be best. As of a result of these concerns I must recommend rejection of the paper.

Response 1: Thank you very much for your valuable suggestions. We are happy to implement the changes you proposed in your second suggestion. However, as it is currently the middle of the undergraduate summer vacation, we are unable to collect new data under similar conditions as in Experiment 2. Given the short timeframe for revisions requested by the journal, we apologize that we will have to follow your first suggestion to improve the rigor of this study.

As you suggested, we have made the following changes:

First, we added the following statement to the last paragraph of section 4.2 of the General Discussion: "It is important to note, however, that in Experiment 2, the results of the ANOVA on the JOLs indicated that the interaction between belief and encoding type did not reach statistical significance. The findings that metamemory beliefs about encoding styles affected participants' JOLs were obtained through post-hoc comparisons. Consequently, these results need to be treated with statistical caution." (Pages 10, lines 415-419).

Second, we have revised the last paragraph of section 4.3 of the General Discussion, which now reads as follows: "Finally, there are Third obvious limitations of this study. First, as mentioned above, although Experiment 2 found that participants’ metamemory beliefs affected their JOLs, this result was obtained from a simple effects analysis conducted after the interaction was found to be non-significant. Therefore, future experimental validation is needed to ensure the reproducibility of this result and to draw scientifically rigorous conclusions. Second, although this study demonstrated that beliefs do play a role in the effect of encoding strategies on JOLs, only one validation method (belief manipulation) was used. Other methods include belief questionnaires [25,26], pre-study JOLs [25,31], and learner-observer tasks [50,51]. Given that each method has its strengths and limitations [35], different measures should be used in the future to verify the reliability of our results. Third, this study did not isolate the role of processing fluency. Furthermore, the effect of encoding style on JOLs may also be influenced by factors such as the level of effort during encoding and level of mental arousal. Therefore, it has yet to be verified whether beliefs play a unique role herein.". (Pages 10-11, lines 447-460)

Comments 2: The error bars are very large relative to the difference between conditions, and in fact it’s difficult to believe that there are any significant differences between JOLs in Figure 3 (the error bars are almost entirely overlapping). I’m guessing the error bars are raw standard deviations, which are not particularly helpful when visually assessing significance. I’d highly recommend they use confidence intervals instead, preferably with a within-subject correction (e.g. Loftus-Masson). In any case, the authors need to specify what the error bars are in the figure caption.

Response 2: Thank you for pointing this out! Previously, the error bars represented raw standard deviations. Based on your suggestion, we have changed them to confidence intervals with a within-subject correction (Figures 2 and 3). We have also updated the figure captions to reflect this change with the sentence, "The error bars are confidence intervals" (Page 6, line 235, and Page 8, lines 316-317).

Comments 3: The authors say for both recall performance and JOLs that they performed repeated-measures ANOVAs (lines 306, 320), which would mean that both factors were within-subject, when they should have performed mixed ANOVAs with belief as a between-subject factor and encoding type within-subject. Presumably this was a labeling error and the authors did not treat both variables as within-subject.

Response 3: Thank you very much for pointing out this labeling error. We have corrected it by changing "repeated-measures ANOVAs" to "mixed ANOVAs" (Page 7, line 305, and Page 8, line 320).

Comments 4: The 1 and 2 footnotes for Table 1 should be switched, I believe (SEM should correspond to semantic encoding).

Response 4: We apologize for the mislabeling of the footnote due to our carelessness. We have corrected it (Page 5, line 227).

Comments 5: I was a little confused in the abstract when the authors called the self-reference effect a memory monitoring process (lines 8-9), but it seems to me that the effect itself doesn’t necessarily have anything to do with monitoring per se (although the authors are investigating monitoring in the context of effects of self-referencing on metamemory).

Response 5: Thank you for pointing out the error. There was a writing mistake; the self-referencing effect is not a "memory monitoring process", and we have removed that part. The sentence now reads: "However, the question arises as to whether this effect could extend to higher cognitive domains such as metamemory." (Page 1, lines 8-9). We apologize again for our carelessness.

Comments 6: It seems odd that the power analyses of Experiments 1 and 2 would calculate that the exact same number of participants (34; lines 175 and 270) was needed for both experiments given that Experiment 2 had a between-subjects manipulation.

Response 6: Thank you for pointing this out. We reviewed the calculation process for Experiment 1 (with power = 0.80, α = 0.05, and a medium effect size of 0.5) and Experiment 2 (with power = 0.80, α = 0.05, and a medium effect size of 0.25) using the G*Power 3.1.9.7 statistical software. We have confirmed that the previously calculated sample sizes for both Experiment 1 and Experiment 2 were correct. Regarding Experiment 2, we were particularly cautious about the sample size provided by the software. Consequently, we recruited 60 participants (30 participants for each of the two belief groups) to achieve a larger sample size and enhance the reliability of our results.

Round 2

Reviewer 2 Report

Comments and Suggestions for Authors

I thank the authors for addressing my concerns. Although my statistical concerns still stand, it is certainly an improvement that the they are now acknowledged in the paper, as is the need for more research to determine if this effect is reliable. I am fine with publishing the new draft of the paper.

Author Response

Thank you very much for your thoughtful comments and for taking the time to review our revised manuscript. We will continue to explore this area in future studies to determine the reliability of the observed effect. Your feedback has been invaluable in enhancing the quality of our work, and we are grateful for your support in moving forward with the publication.